# Comparative Analysis of the Chloroplast Genomes of Eight Species of the Genus *Lirianthe* Spach with Its Generic Delimitation Implications

**DOI:** 10.3390/ijms25063506

**Published:** 2024-03-20

**Authors:** Tao Wu, Yong-Kang Sima, Shao-Yu Chen, Yu-Pin Fu, Hui-Fen Ma, Jia-Bo Hao, Yun-Feng Zhu

**Affiliations:** Yunnan Key Laboratory of Biodiversity of Gaoligong Mountain, Yunnan Academy of Forestry and Grassland, Kunming 650201, China; ynafwt@126.com (T.W.); sherry9872@163.com (S.-Y.C.); fuyupin@yafg.ac.cn (Y.-P.F.); mahf_2004@126.com (H.-F.M.); jj2000ms@163.com (J.-B.H.); kji-888@163.com (Y.-F.Z.)

**Keywords:** Magnoliaceae, *Lirianthe* Spach, chloroplast genome, phylogeny

## Abstract

Based on Sima and Lu’s system of the family Magnoliaceae, the genus *Lirianthe* Spach s. l. includes approximately 25 species, each with exceptional landscaping and horticultural or medical worth. Many of these plants are considered rare and are protected due to their endangered status. The limited knowledge of species within this genus and the absence of research on its chloroplast genome have greatly impeded studies on the relationship between its evolution and systematics. In this study, the chloroplast genomes of eight species from the genus *Lirianthe* were sequenced and analyzed, and their phylogenetic relationships with other genera of the family Magnoliaceae were also elucidated. The results showed that the chloroplast genome sizes of the eight *Lirianthe* species ranged from 159,548 to 159,833 bp. The genomes consisted of a large single-copy region, a small single-copy region, and a pair of inverted repeat sequences. The GC content was very similar across species. Gene annotation revealed that the chloroplast genomes contained 85 protein-coding genes, 37 tRNA genes, and 8 rRNA genes, totaling 130 genes. Codon usage analysis indicated that codon usage was highly conserved among the eight *Lirianthe* species. Repeat sequence analysis identified 42–49 microsatellite sequences, 16–18 tandem repeats, and 50 dispersed repeats, with microsatellite sequences being predominantly single-nucleotide repeats. DNA polymorphism analysis revealed 10 highly variable regions located in the large single-copy and small single-copy regions, among which *rpl32-trnL*, *petA-psbJ*, and *trnH-psbA* were the recommended candidate DNA barcodes for the genus *Lirianthe* species. The inverted repeat boundary regions show little variation between species and are generally conserved. The result of phylogenetic analysis confirmed that the genus *Lirianthe* s. l. is a monophyletic taxon and the most affinal to the genera, *Talauma* and *Dugandiodendron*, in Sima and Lu’s system and revealed that the genus *Lirianthe* s. s. is paraphyletic and the genus *Talauma* s. l. polyphyletic in Xia’s system, while *Magnolia* subsection *Gwillimia* is paraphyletic and subsection *Blumiana* polyphyletic in Figlar and Nooteboom’s system. Morphological studies found noticeable differences between *Lirianthe* species in aspects including leaf indumentum, stipule scars, floral orientation, tepal number, tepal texture, and fruit dehiscence. In summary, this study elucidated the chloroplast genome evolution within *Lirianthe* and laid a foundation for further systematic and taxonomic research on this genus.

## 1. Introduction

The family Magnoliaceae encompasses a group of plants that are highly valued for their ornamental qualities. The family involves about 300 species and ranges across tropical and temperate areas, mainly in Southeast Asia, North South America Central America, Southeast North America, including Mexico, and Antilles [1,2]. In terms of classification, dividing Magnoliaceae into two subfamilies or two tribes, Liriodendroideae (or Liriodendreae) and Magnolioideae (or Magnolieae), has been widely accepted [3,4,5]. Within the subfamily Liriodendroideae, there is no dispute that it contains only the single genus, *Liriodendron* Linnaeus. Currently, all research indicates that the classification controversies within the family Magnoliaceae are mainly focused on the delimitation and classification of genera within the subfamily Magnolioideae. The number of genera within the subfamily Magnolioideae has been divided into 16 by Xia [6], 15 by Law [4], 14 by Sima and Lu [7], 11 by Dandy [3], or even just 1 by Figlar and Nooteboom [8]. Li [9] prefer the treatment of segregated or smaller genera in the family Magnoliaceae because that the treatment of the bigger genera cannot represent the later evolution process and levels in this family, and that the bigger genera are too inclusive to reflect the evolutional trends and the migratory routes mainly on the basis of morpho-geographical studies. Sima and Lu’s Magnoliaceae system [7,10], established in 2012 based on DNA data and morphological characteristics, especially observations on living plants, consists of 14 monophyletic genera in their delimitation of the subfamily Magnolioideae, which has been strongly supported by an increasing number of DNA data analysis results and gradually accepted by many scholars [11,12,13,14].

The genus *Lirianthe* Spach s. l. in Sima and Lu’s system is equivalent to *Magnolia* subgenus *Magnolia* section *Gwillimia* subsection *Gwillimia* Candolle plus the subsection *Blumiana* (Blume) Figlar & Nooteboom in Figlar and Nooteboom’s system [15]. It is a monophyletic group [7,10] and includes about 25 species mainly distributed from East Himalaya through South China to Southeast Asia [16]. There are about 12 species of this genus in China, of which 5 are endemic to China [16,17]. But this genus *Lirianthe* Spach s. s. in Xia’s system excludes those species with the circumscissile mature carpels such as *L. hodgsonii* (J. D. Hooker & Thomson) Sima & S. G. Lu, and so on. Plants of the genus *Lirianthe* have very important practical value and are world-renowned ornamental plants, industrial timber sources, medicinal materials, and fragrance source species [18,19]. *L. henryi* (Dunn) N. H. Xia & C. Y. Wu, *L. coco* (Loureiro) N. H. Xia & C. Y. Wu, *L. delavayi* (Franchet) N. H. Xia & C. Y. Wu and *L. odoratissima* (Y. W. Law & R. Z. Zhou) N. H. Xia & C. Y. Wu have large and fragrant flowers, making them popular choices for garden ornamentals. Additionally, their lush green foliage makes them ideal for greening up outdoor space. *L. phanerophlebia* (B. L. Chen) Sima is only known from Hekou, Jinping, and Maguan in the south-east of Yunnan province. Based on the results of the Global Trees Campaign field surveys in December 2005, it is estimated that the total wild population is less than 200 individuals. The biggest threat to the species is a decrease in habitat, with many suitable areas now replaced by banana plantations. Local awareness-raising is vital for this species, as is research into nursery techniques for its cultivation. *L. henryi* is a national second-level key protected wild plant from Yunnan province. It has been listed as an endangered plant by the international union for conservation of nature (IUCN). *L. confusa* Sima & W. N. Sima and *L. brevisericea* Sima & Hong Yu are two newly described species.

Earlier phylogenetics of Magnoliaceae plants based on DNA analysis employed from one to several nuclear or chloroplast DNA regions [20,21,22], but a recent systematic and phylogenetic study based on complete chloroplast genome (CPG) sequences of nine species of *Lirianthe* demonstrated that these nine species of *Lirianthe* form a monophyletic branch [23]. This study shows the value of chloroplast genomic data for systematic or species classification research, but did not include *L. henryi*, *L. phanerophlebia*, *L. confusa*, and *L. brevisericea*. Here, the complete chloroplast genomes were sequenced and assembled for eight species of genus *Lirianthe*. Our goal was to (1) elucidate the genomic structure, gene content and genetic variability encoded in *Lirianthe* chloroplast DNA; (2) compare the chloroplast phylogenomics in the family Magnoliaceae using CPG sequences and protein coding sequence separately, with an emphasis on the generic delimitation for *Lirianthe*. The chloroplast sequence datasets established herein facilitate improved phylogenetic resolution and evolutionary inferences for these under-surveyed Asian endemics and their underexplored *Lirianthe* relatives.

## 2. Results

### 2.1. The Sequence Coverage Analysis of Assembled Chloroplast Genome

The sequence coverage, also known as the sequencing depth, serves as a key indicator for determining the reliability and the quality of assembly results. The average sequence coverages of eight assembled plastomes ranged from 93.9× to 547.9×, evaluated by GetOrganelle. The raw sequencing reads were also aligned to the assembled plastomes using minimap2 and the coverage depth was calculated using samtools, which ranged from 105.79× to 1577.05×. The sequencing coverages are shown in Table 1. The specific coverage results of eight chloroplast assemblies, obtained through the minimaup2 + samtools methods, can be found in Appendix A, and their graphical results are illustrated in Appendix A.

### 2.2. General Characteristics of Eight Chloroplast Genomes of Lirianthe Species

The sizes of the CPGs of eight *Lirianthe* species ranged from 159,548 bp to 159,833 bp, with significant variations between individual species (Table 2). The nucleotide compositions of the genomes were remarkably similar to each other, with 30.0% Adenine (A), 30.7% Thymine (T), 20.0% Cytosine (C), and 19.3% Guanine (G). This suggests a close evolutionary relationship among them. The CPGs were divided into a canonical quadripartite structure with a large single-copy region (LSC, 87,671~87,965 bp), a small single-copy region (SSC, 18,745~18,772 bp), and a pair of inverted-repeats regions (IRa/IRb, 26,538~26,560 bp). Figure 1 shows the chloroplast genome structure of the genus *Lirianthe*, taking *L. confusa* as an example. The GC content of the CPG, LSC, SSC, and IR regions were 39.27%~39.29%, 37.97%~37.98%, 34.32%~34.46%, and 43.16%~43.18%, respectively (Table 2). The number of annotated genes in the eight chloroplast genomes is the same, with a total of 130, including 85 protein-coding genes, 37 transfer RNA genes, and 8 ribosomal RNA genes. The gene names and classifications of the 85 protein-coding genes were shown in Appendix A, and were classified into four categories for their different roles. The annotated CPG data have been submitted to the NCBI database. The specific accession numbers are shown in Table 2.

### 2.3. Codon Usage of Protein-Coding Genes

Among the 85 protein-coding genes (PCGs), after removing 1 of the 7 genes with 2 copies (*ndhB*, *rpl2*, *rpl23*, *rps7*, *rps12*, *ycf2*, *ycf15*), the remaining 78 PCGs were used for codon usage analysis.

The relative synonymous codon usage (RSCU) values of eight *Lirianthe* CPGs are displayed in Appendix A.

Perhaps due to the high genetic similarity, the RSCU values of the eight species are very close, and the RSCU analysis results of *L. confusa* (MT654129) are shown in Figure 2. Its protein-coding gene region contained 22,659 codons (including 78 terminal codons). Among these codons, Leu (Leucme, L) was the most abundant amino acid, with 2314 (10.21%), while Cys (Cysteine, C) was the least, with 262 (1.16%). In all the termination codons, AUU is the most frequently used amino acid.

Except for the stop codon, there are 30 codons with RSCU values greater than one, in which the codon ending with A/U was 94.80% and the codon ending with C/G was 5.20%, indicating that these codons tended to end in A/U. The codon usage of these *Lirianthe* species is notably conserved, and is consistent with earlier reports on the chloroplast genomes of many land plants [24].

### 2.4. Repeat Identification

A total of 42~49 microsatellite sequence repeats (SSR) loci, 16~18 tandem repeats, and 50 dispersed repeats were detected in the eight *Lirianthe* CPGs (Table 3). The SSRs of eight species were all single nucleotides and dinucleotides, and single nucleotides were dominant, accounting for more than 90%; more than three nucleotide repeating units were not detected. Among the dispersed repeat sequences, palindromic repeats and forward repeats were the main ones, accounting for 50% and 30%, respectively.

### 2.5. DNA Polymorphism

The polymorphism of the CPG nucleotide sequence is displayed graphically using the nucleotide diversity values (Pi) obtained by sliding window analysis, as shown in Figure 3. This allowed for a comprehensive visualization of the genetic variations among CPGs. The Pi value ranges from 0 to 0.012, with an average of 0.0014. The 10 intervals with the highest Pi values are shown in Figure 3, and are *petA-psbJ*, *trnH-psbA*, *psbM-trnY*, *rpl32-trnL*, *ccsA-ndhD*, *accD-ycf4*, *ndhF*, *ycf1*, *matK-rps16*, and *trnN-ndhF*, respectively. Moreover, these highly variable regions are in both the LSC and SSC regions. The *petA-psbJ* has the highest Pi value (0.012). Furthermore, only the LSC and SSC regions include these highly variable regions; this finding is also supported by the graphical results of sequence alignments (Appendix A).

### 2.6. Contraction and Expansion of IR Region

The length of the IR region of the eight *Lirianthe* species varies between 26,538 and 26,560 bp (Table 2). To better understand the boundary and detect changes in genes, we employed IRscope software (https://irscope.shinyapps.io/irapp/) to visualize the chloroplast into four regions: LSC-IRb (JLB), IRb-SSC (JSB), SSC-IRa (JSA), and IRa-LSC (JLA). Although the differences among them are small, the IR boundary region does show some differences in Appendix A.

### 2.7. Phylogenetic Analysis

The phylogenetic tree was constructed for the 8 *Lirianthe* species and the other 36 Magnoliaceous species using maximum likelihood (ML) and Bayesian methods based on the complete chloroplast genome (CPG) data. The ML consensus tree (Figure 4) was congruent with the Bayesian one (Figure 5) in the phylogenetic topologies, showing that the 44 Magnoliaceae species were divided into 15 monophyletic groups (*Aromadendron*, *Dugandiodendron*, *Houpoea*, *Kmeria*, *Lirianthe*, *Liriodendron*, *Magnolia*, *Manglietia*, *Metamagnolia*, *Michelia*, *Oyama*, *Pachylarnax*, *Paramagnolia*, *Talauma*, *Yulania*), successfully supporting Sima and Lu’s system [7]. But the 15 species of the genus *Lirianthe* had been scattered on different branches and classified in the two different genera, *Talauma* and *Lirianthe*, in Xia’s system [6] or in the two different subsections, *Magnolia* subsection *Gwillimia* and subsection *Blumiana*, in Figlar and Nooteboom’s system [8]. The statistical supports, bootstrap values (BP) and posterior probabilities (PP), varied slightly among some branches or clades in Figure 4 and Figure 5, but overall were relatively high, indicating that the tree-building results from ML and Bayesian methods both achieved good results in this study.

### 2.8. Plant Morphology

The main morphological characteristics of the leaves, flowers, and fruits of the eight *Lirianthe* species are described in Table 4. Except for the midveins of the leaf blade of *L. henryi* and *L. hodgsonii*, which are adaxially prominent, the other six species are all impressed. Regarding the presence and texture of trichomes, *L. brevisericea*, *L. confusa*, *L. delavayi* and *L. odoratissima* have various types of indumenta on the abaxial leaf surface, including sericeous and tomentose hairs; *L. henryi* has scattered appressed trichomes on the abaxial surface; whereas *L. coco*, *L. hodgsonii* and *L. phanerophlebia* have glabrous leaves. Except for *L. phanerophlebia*, whose stipule scars only extend to 1/3–1/2 of the petiole length, all other species have stipule scars that reach the top of the petiole. In terms of floral orientation, except for *L. brevisericea*, *L. delavayi*, and *L. hodgsonii* which are erect, the flowers of the remaining species are pendulous. The tepals are mostly 9 in number, ranging from 8 to 12; they are usually white, with only *L. delavayi* sometimes pink or red. The presence, texture, sparseness, and color of indumentum on the gynoecia are key distinguishing characteristics. Except for the mature carpel of *L. hodgsonii*, which dehisces along a circumference, all other species dehisce along the dorsal suture.

## 3. Discussion

The chloroplast genome is a superior option to the nuclear genome for studying nucleotide diversity and reconstructing the phylogeny of related species due to its smaller genome size, lower nucleotide substitution rate, uniparental inheritance, and haploid characteristics [25]. The development of next-generation sequencing technology and bioinformatics analysis methods has significantly reduced the cost of obtaining genome sequences. Therefore, chloroplast genome-scale data are seeing increasing use. As a result, inferring evolutionary relationships at higher taxonomic levels [26], even at the interspecific or intraspecific level in some species, such as ginseng authentication by 18 species-specific markers [27], authentication of ginseng cultivars by 17 polymorphic sites [28], differentiation of hazelnut cultivars by a combination of markers of two InDels and one SNP [29], markers were mostly developed from whole-chloroplast genomes in these studies. The chloroplast genome assembly and annotation, as well as comparative analysis, are the basis and prerequisite for the above work.

We had previously completed the assembly and annotation of chloroplasts of *L. hodgsonii* [23] and *L. coco* [30]. Here, we newly assembled the chloroplast genomes of six other *Lirianthe* species. The CPGs of these plants were found to possess a similar size and structure, all demonstrating a typical quadripartite pattern. There is a known connection between the phylogenetic position and the total GC content. Early differentiated lineages, like those found in the Magnoliaceae, have a greater GC content [31]. Compared to the median GC content of 35% in most angiosperms, plants of the genus *Houpoea* N. H. Xia & C. Y. Wu have a higher GC content of around 39.2% [32], which is almost identical to the GC content of 39.3% in the *Lirianthe* species in this study. The GC content in the IR region of all *Lirianthe* species is notably high (43.2%), and alignment is increasingly conducted using chloroplast genome-scale data, with similar findings in other plants like *Carthamus* (43.2%) [33] and *Cypripedium* (42.7%) [34], potentially due to the inclusion of four highly conserved rRNA genes with high GC content (Figure 1).

SSR markers have been widely used in population genetics research due to their reliability and high variability. In this study, a small amount of SSR markers were detected, with about 170 in *Houpoea* species [32] and 246 in *Bougainvillea spectabilis* [35], while our findings indicated only 42~49 SSRs. The primary explanation is the use of different search parameters, as *Carthamus* species can only recognize 36–40 SSRs [33] when the same parameters (1–10 2–6 3–5 4–5 5–5 6–5) are applied. The expansion and contraction of IR regions at the junctions of LSC and SSC have significant impacts, including the creation of pseudogenes and alterations in genome size and evolutionary rate [36]. There was an absence of substantial progress even at lower taxonomic levels. Species identification and the detection of structural changes in IR boundaries in this study suggest that Magnoliaceae may have remained relatively primitive and conservative in nature.

The analysis of CPG DNA polymorphism is a proven approach to detecting mutation hotspots, which can function as distinct DNA barcodes. Two of the ten genes or DNA segments with the highest assessment of nucleotide diversity detected in this study, *rpl32-trnL* and *petA-psbJ*, are also present in the genus *Houpoea* [32], which also belongs to the Magnoliaceae family. *trnH-psbA* is one of the early recommended plant DNA barcodes [37] and was also the second most polymorphic fragment in this study. *rpl32-trnL*, *petA-psbJ*, and *trnH-psbA* are good candidate barcodes for *Lirianthe* species based on the results of comparative analysis of DNA polymorphisms. These potentially highly variable chloroplast barcodes will greatly enhance our ability to identify and protect rare and endangered species within the genus *Lirianthe*. Resolution of phylogenetic relationships and reconstruction of evolutionary history have become feasible through comparison of chloroplast genomes and phylogenetic analysis.

The genus *Lirianthe* s. s. is one of the genera reinstated by Xia and Wu for the *Magnolia* subsection *Gwillimia* when several genera were recognized in the family Magnoliaceae. It has been accepted in many recent publications from China and Vietnam [38]. CPG has proved to be a powerful tool in unraveling the evolutionary associations among land plants [26,39,40]. Based on Sima and Lu’s system [7], 15 species of the genus *Lirianthe* s. l. and 29 species of other genera of Magnoliaceae were selected, and a phylogenetic tree was drawn using a ML method with chloroplast whole-genome sequences (Figure 4). Among them, all 15 species of the genus *Lirianthe* s. l. gathered together and formed an independent branch. This confirmed that the genus *Lirianthe* s. l. is a monophyletic taxon in Sima and Lu’s system. But the 15 species of the genus *Lirianthe* s. l had been scattered on different branches and classified in the two different genera, *Talauma* and *Lirianthe*, in Xia’s system [6] or in the two different subsections, *Magnolia* subsection *Gwillimia*, and subsection *Blumiana* in Figlar and Nooteboom’s system [8]. These revealed that the genus *Lirianthe* s. s. is paraphyletic and the genus *Talauma* s. l. polyphyletic in Xia’s system, while the *Magnolia* subsection *Gwillimia* is paraphyletic and the subsection *Blumiana* is polyphyletic in Figlar and Nooteboom’s system. Furthermore, the 15 *Lirianthe* species show a crossover mixed pattern in Xia’s system. The four species marked with green shading demonstrate the results of this cross-mixing classification in Figure 4. The processing results of *Lirianthe* plants indicate that the plant range of *Lirianthe* delimited in Sima and Lu’s system is reasonable and scientific, and chloroplast genome evidence supports these taxonomic treatments.

However, as mentioned in the introduction, there are too many generic delimitation problems within the tribe Magnolieae or the subfamily Magnolioideae. In 2004, Figlar and Nooteboom, based on the latest available data on DNA and morphology, degraded the tribe Magnolieae or the subfamily Magnolioideae into the genus *Magnolia* Linnaeus, reduced all of its former segregated genera to *Magnolia*, and reconstructed a complex infrageneric system of the biggest genus, *Magnolia* s.l., in the family Magnoliaceae, including 3 subgenera, 12 sections, and 13 subsections. Thus, it appeared that this treatment had solved the generic delimitation problems within the tribe Magnolieae or the subfamily Magnolioideae. In essence, this just changed the problems from generic delimitation to infrageneric delimitation and the delimitation problems remained unsolved. In order to better unravel the generic delimitation problems above, Xia [6] and Sima & Lu [7] published a new system for the family Magnoliaceae in 2012. A total of 17 genera are recognized in Xia’s system [6] and a total of 15 genera are recognized in Sima and Lu’s system [7]. The system by Sima and Lu, based on the data on DNA and the observations of morphological characters, especially in living plants, is strongly supported by recent DNA work [21,23,30,32,41] and has been accepted by many scholars [11,12,13,14]. In order to better reflect the evolutionary steps and levels of Magnoliaceous plants, as well as the evolutionary trends and migration routes based primarily on morphological–geographical studies, it is a more feasible and useful approach to define smaller independent genera. Overall, the results of this study can provide valuable sequence information for the molecular systematics of *Lirianthe* in Magnoliaceae and offer a theoretical basis for the utilization and conservation of *Lirianthe* germplasm resources.

## 4. Materials and Methods

### 4.1. Sample DNA Extraction and Sequencing

We dried and preserved the collected fresh leaves of 8 species (*L. brevisericea*, *L. coco*, *L. confuse*, *L. delavayi*, *L. henryi*, *L. hodgsonii*, *L. odoratissima*, *L. phanerophlebia*) with silica gel in 2019 and 2020. The voucher specimens were deposited at the herbaria of YAF and YCP, Kunming City, China (Y.-K. Sima & S.-Y. Chen 99,313 for *L. brevisericea*; Y.-K. Sima & S.-Y. Chen 99,279 for *L. coco*; Y.-K. Sima 99,261 for *L. confus*; Y.-K. Sima & S.-Y.Chen 99,277 for *L. delavayi*; Y.-K. Sima & S.-Y. Chen 99,315 for *L. henryi*; H. Jiang 7337 for *L. hodgsonii*; Y.-K. Sima 99,263 for *L. odoratissima*; Y.-K. Sima & Y.-M. Shui 99,321 for *L. phanerophlebia*). We extracted the total genomic DNA from the samples using DNA Plantzol Reagent (Invitrogen, Carlsbad, CA, USA), and sent the DNA to the BGI Group (Shenzhen, China) for library construction and sequencing on an HiSeq 2500 platform (Illumina, Hayward, CA, USA). The 2 × 150 bp paired-end reads with an insert size of 400 bp were created and the resulting DNA sequences were obtained after removing reads with low quality and adapter contamination.

### 4.2. Chloroplast Genome Assembly and Annotation

The raw sequencing reads were assembled by the programs GetOrganelle v1.7.7.0 using default parameters [42]. The genome annotation was conducted by using CPGAVAS2 [43], with manual correction using Geneious software (version 2020.0.5) [44]. The chloroplast genome coverage analysis was evaluated in two ways. The first was based on the assembly log from GetOrganelle. In the second evaluation, the raw reads were mapped to the assembled genomes using minimap2, and then samtools was used to count the read coverage at each position of the genome.

### 4.3. The Analysis of Structures and Characteristics of CPGs

The general statistics of the CPGs, including genome size, number of genes, gene size, nucleotide composition, and so on, were evaluated using the Geneious software [44]. The structure map was also drawn using Geneious [44]. The identification of three types of repeats, namely microsatelite sequence repeats (SSRs), tandem repeats and dispersed repeats, was calculated using CPGAVAS2 [43]. The parameters were set as follows: 1–10 2–6 3–5 4–5 5–5 6–5 for SSRs, 2 7 7 80 10 50 500 −f −d −m for tandem repeats, −f −p −h 3 −l 30 for dispersed repeats. The DNA sequences of protein-coding genes (PCGs) were extracted using PhyloSuite v1.2.3 [45], then the frequency of relative synonymous codon usage (RSCU) was calculated using MEGA v11 [46].

### 4.4. Comparitive Analysis of Chloroplast Genomes

To check the polymorphism of DNA, the CPGs sequences of 8 *Lirianthe* species were aligned using MAFFT v7.490 [47], then sliding window analysis was performed on the alignment result using DnaSP v5.10.1 [48] with the window length and step size both set to 500 bp. The regions with high Pi values were considered as mutational hotspots or candidate DNA barcodes. The contraction and expansion of IR region, namely LSC/IRb/SSC/IRa junctions, can be visualized in the online program IRscope (https://irscope.shinyapps.io/irapp/) [49].

### 4.5. Phylogenetic Analysis

To compare the phylogenetic positions of the genus *Lirianthe* in the family Magnoliaceae, 36 chloroplast genome sequences of other *Lirianthe* species and other genera in Magnoliaceae were downloaded from the NCBI database. All 44 species of plants from the Magnoliaceae family covered in this study and their accession number of the chloroplast genome are listed in Appendix A. The phylogenetic trees were built using the complete chloroplast genome (CPG) DNA sequences of 44 species by maximum likelihood (ML) and Bayesian methods. The CPG DNA was aligned using MAFFT [47] in Geneious [44] with default settings. The ML analysis was run by RAxML 8.2.11 program [50] under the GTR + GAMMA model with 1000 bootstrap replicates. The Bayesian analysis was run by MrBayes 3.2.6 program [51] under the GTR + GAMMA model with Markov Chain Monte Carlo (MCMC) settings of 2 runs of 1,000,000 generations with a burn-in of 250,000 generations with trees sampled every 1000 runs. The phylogenetic tree was visualized by iTOL v5 (https://itol.embl.de/) [52].

## 5. Conclusions

By studying the chloroplast genome of eight *Lirianthe* species, we were able to clearly outline their characteristics and provide valuable insights into the divergence and phylogenetic evolution of *Lirianthe* species. It was evident from our analysis that the *Lirianthe* species exhibited a strong level of conservation in its genome structure, gene content, IR boundaries, repetitive elements, and codon usage. This implies that these species have maintained a steady genetic structure throughout their evolutionary history. The chloroplast genome is a robust tool for unraveling the phylogenetic connections among species within the genus *Lirianthe*, as shown by the high resolution of the ML tree. Our investigation of phylogenetic relationships has provided valuable insights into the connections between *Lirianthe* and other genera within family Magnoliaceae. The knowledge presented here enhances our understanding of how these species are related. By enriching the knowledge of the chloroplast genome of *Lirianthe* species, our study paves the way for future investigations into their classification and evolution.

## Figures and Tables

**Figure 1 ijms-25-03506-f001:**
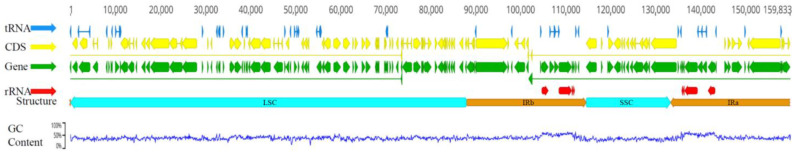
The gene and structure diagram of the *Lirianthe* species chloroplast genomes, taking *L. confusa* (MT654129) as an example.

**Figure 2 ijms-25-03506-f002:**
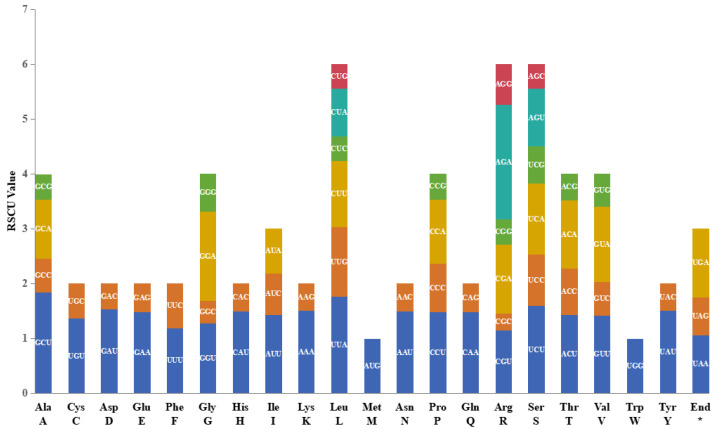
Relative synonymous codon usage (RSCU) analysis, taking *L. confusa* (MT654129) as an example. The asterisk (*) indicates terminator codon.

**Figure 3 ijms-25-03506-f003:**
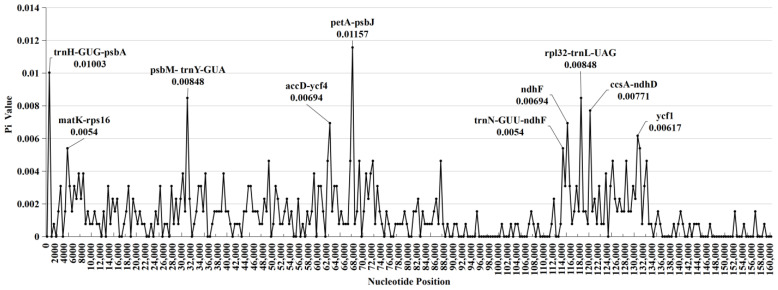
The DNA polymorphism analysis of eight *Lirianthe* chloroplast genomes by sliding window method. Window length: 500 bp; step size: 500 bp.

**Figure 4 ijms-25-03506-f004:**
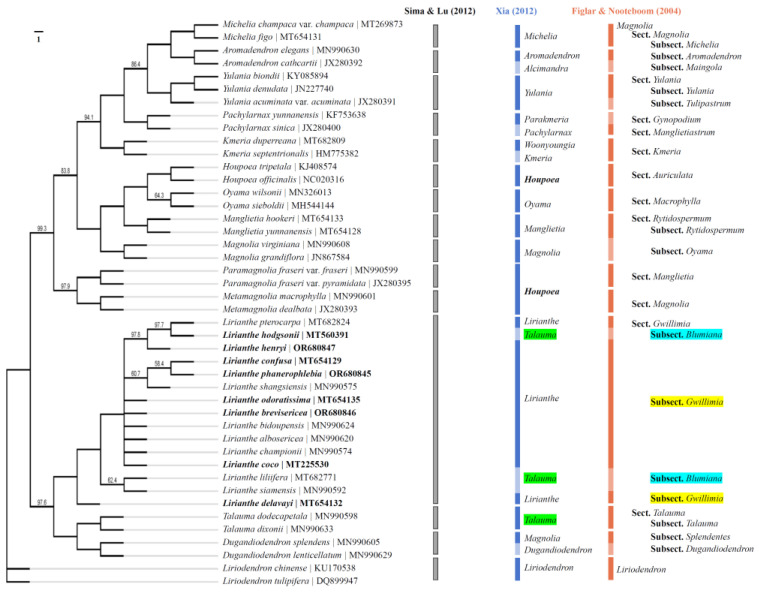
Phylogenetic trees of the genus *Lirianthe* and other Magnoliaceous species based on whole-chloroplast genome sequences using Maximum likelihood method based on 1000 replicates. Bootstrap percentages less than 100 are shown. Note: Gray, blue and orange bars indicate the results or belonging positions of the corresponding plants in Sima and Lu’s [7], Xia’s [6] and Figlar and Nooteboom’s [8] systems, respectively. The green background color highlights the genus *Lirianthe* had been scattered on different branches and classified in the two different genera, *Talauma* and *Lirianthe*, in Xia’s system [6]; The blue and yellow background colors highlight the genus *Lirianthe* had been scattered in the two different subsections, *Magnolia* subsection *Gwillimia*, and subsection *Blumiana* in Figlar and Nooteboom’s system [8].

**Figure 5 ijms-25-03506-f005:**
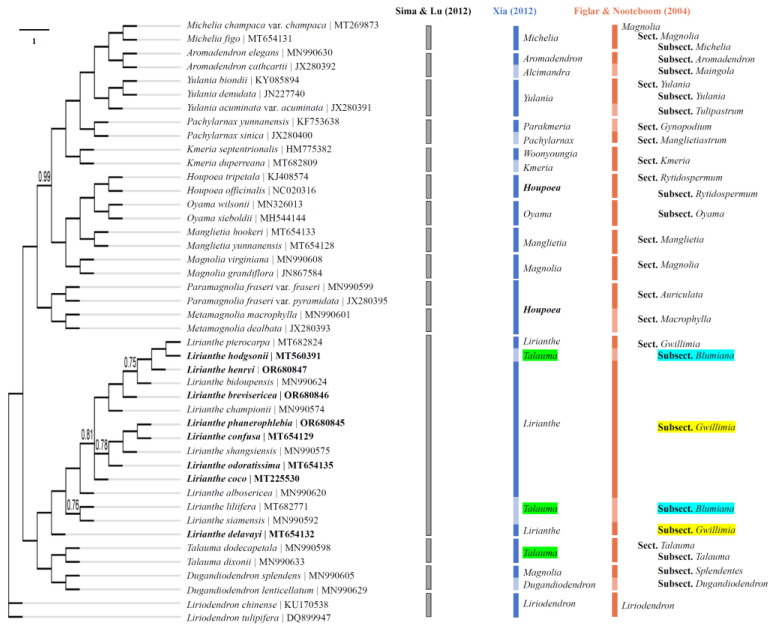
Phylogenetic trees of the genus *Lirianthe* and other Magnoliaceous species based on whole-chloroplast genome sequences using Bayesian method. Numbers above branches are bootstrap support values based on 1,000,000 generations with a burnin proportion of 0.25. Posterior probabilities less than 1 are shown. Note: Gray, blue, orange bars and different background colors are same as Figure 4.

**Table 1 ijms-25-03506-t001:** The assembly coverages of eight *Lirianthe* chloroplast genomes evaluated using two methods (average depth, ×).

Methods	*L. brevisericea*	*L. coco*	*L. confusa*	*L. delavayi*	*L. henryi*	*L. hodgsonii*	*L. odoratissima*	*L. phanerophlebia*
GetOrganelle	93.9	475.9	483.3	547.3	374.1	328.2	547.9	236.0
Minimap + samtools	105.79	845.20	1305.89	368.77	1577.05	300.42	1147.52	273.37

**Table 2 ijms-25-03506-t002:** Features of eight *Lirianthe* chloroplast genomes.

Species	*L. brevisericea*	*L. coco*	*L. confusa*	*L. delavayi*	*L. henryi*	*L. hodgsonii*	*L. odoratissima*	*L. phanerophlebia*
Name Abbr.	Lbre	Lcoc	Lcon	Ldel	Lhen	Lhod	Lodo	Lpha
Accession No.	OR680846	MT225530	MT654129	MT654132	OR680847	MT560391	MT654135	OR680845
Total Length (bp)	159,811	159,828	159,833	159,714	159,760	159,693	159,819	159,548
LSC (bp)	87,933	87,958	87,965	87,877	87,891	87,846	87,961	87,671
SSC (bp)	18,758	18,760	18,752	18,761	18,757	18,745	18,772	18,761
IR (bp)	26,560	26,555	26,558	26,538	26,556	26,551	26,543	26,558
Total Genes	130							
CDS	85							
tRNA	37							
rRNA	8							
Total GC%	39.28	39.28	39.27	39.29	39.28	39.28	39.27	39.29
LSC (GC%)	37.98	37.97	37.97	37.97	37.97	37.97	37.97	37.98
SSC (GC%)	34.36	34.37	34.39	34.46	34.38	34.37	34.32	34.41
IR (GC%)	43.17	43.18	43.16	43.18	43.18	43.18	43.18	43.17
A (%)	47,908 (29.98)	47,915 (29.98)	47,913 (29.98)	47,862 (29.97)	47,868 (29.96)	47,851 (29.96)	47,908 (29.98)	47,799 (29.96)
C (%)	31,980 (20.01)	31,983 (20.01)	31,979 (20.01)	31,972 (20.02)	31,980 (20.02)	31,958 (20.01)	31,977 (20.01)	31,940 (20.02)
G (%)	30,790 (19.27)	30,795 (19.27)	30,790 (19.26)	30,779 (19.27)	30,777 (19.26)	30,774 (19.27)	30,782 (19.26)	30,741 (19.27)
T (%)	49,133 (30.74)	49,135 (30.74)	49,151 (30.75)	49,101 (30.74)	49,135 (30.76)	49,110 (30.75)	49,152 (30.75)	49,068 (30.75)

**Table 3 ijms-25-03506-t003:** The type and number of repeat sequences in the chloroplast genomes of 8 *Lirianthe* species.

Species		Lbre	Lcoc	Lcon	Ldel	Lhen	Lhod	Lodo	Lpha
SSR	A	17	13	13	17	15	16	16	17
	C	2	2	3	3	3	3	2	3
	G	1	0	1	0	0	1	0	0
	T	21	23	22	25	24	24	21	24
	TA	1	2	2	2	2	2	1	2
	TC	2	2	2	2	2	2	2	2
	Total No.	44	42	43	49	46	48	42	48
Tandem Repeats		16	18	17	17	16	16	17	17
Dispersed Repeats	Complement	0	0	0	0	1	0	0	0
	Forward	16	16	15	18	15	16	17	17
	Palindromic	27	25	26	25	26	26	25	25
	Reverse	7	9	9	7	8	8	8	8
	Total No.	50	50	50	50	50	50	50	50

**Table 4 ijms-25-03506-t004:** Morphological comparisons among eight species of the genus *Lirianthe* Spach.

Species	Leaves	Flowers	Fruits
*L. brevisericea*	Abaxially yellowish-gray sericeous; leaf blade midveins adaxially impressed, lateral veins 12–19 pairs; stipular scars reaching apex of petioles.	Erect; 9 tepals, white; gynoecia densely yellowish-gray sericeous.	Mature carpels dehiscent along dorsal sutures
*L. coco*	Glabrous, leaf blade midveins adaxially impressed, lateral veins 8–16 pairs;; stipular scars reaching apex of petioles.	Pendulous; 9 tepals, white; gynoecia glabrous.	Mature carpels dehiscent along dorsal sutures.
*L. confusa*	Abaxially yellowish-white curved trichomes; leaf blade midveins adaxially impressed, lateral veins 10–15 pairs; stipular scars reaching apex of petioles.	Pendulous; 9 tepals, white; gynoecia very densely yellowish-white villose.	Mature carpels dehiscent along dorsal sutures
*L. delavayi*	Abaxially densely interwoven tomentose and white powdery but later only with residual trichomes on veins; leaf blade midveins adaxially impressed, lateral veins 11–16 pairs; stipular scars reaching apex of petioles.	Erect; 9 to 12 tepals, white, yellowish-white, pink or red; gynoecia fine yellow villose.	Mature carpels dehiscent along dorsal sutures.
*L. henryi*	Abaxially sparsely appressed pubescent; leaf blade midveins adaxially prominent, lateral veins 14–20 pairs; stipular scars reaching apex of petioles.	Pendulous; 9 tepals, white; gynoecia glabrous.	Mature carpels dehiscent along dorsal sutures
*L. hodgsonii*	Glabrous; leaf blade midveins adaxially prominent, lateral veins 10–20 pairs; stipular scars reaching apex of petioles.	Erect; 9 tepals, white; gynoecia glabrous.	Mature carpels circumscissile
*L. odoratissima*	Abaxially yellowish-white or grayish-brown curved trichomes; leaf blade midvein adaxially impressed, lateral veins 9–14 pairs; stipular scars reaching apex of petioles.	Pendulous; 9 to 10 tepals, white; gynoecia densely grayish-brown pubescent.	Mature carpels dehiscent along dorsal sutures
*L. phanerophlebia*	Glabrous; leaf blade midveins adaxially impressed, lateral veins 11–17 pairs; stipular scars 1/3–1/2 as long as petioles.	Pendulous; 8 to 9 tepals, white; gynoecia glabrous or glaucous.	Mature carpels dehiscent along dorsal sutures

## Data Availability

The data provided in this study are deposited in the NCBI GenBank database (https://www.ncbi.nlm.nih.gov/ (accessed on 6 December 2023)); the GenBank accession numbers are listed in Appendix A.

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
