# Peer review of "Comparative Analysis of the Chloroplast Genomes of Eight Species of the Genus *Lirianthe* Spach with Its Generic Delimitation Implications"

_ijms, 2024, doi:10.3390/ijms25063506_

Round 1
Reviewer 1 Report
Comments and Suggestions for Authors This article about phylogeny of the genus Lirianthe uses methods which are typical for chloroplast genome analyses. The methods were used correctly and conclusions are sound. I have only a few minor remarks:- The are typos that should be fixed. "85 protein-coding130 genes" (line 102), "which were classified into fourfive categorie" (line 103), "have been submitted to the NCBI databankdatabas" (line 104).
- In Figure 1, some gene symbols are truncated, like "rp..." or "r...". I think, it is better either to remove gene symbols, or write genes symbols for all 130 genes completely.
- In Figure 1, it is not clear what two green arrows in the rRNA row mean. rRNAs are denoted by red arrows there.
- Please indicate in the caption of Figure 5 that the numbers above branches are bootstrap support values.
- "The chloroplast genome is a superior option to the nuclear genome for studying nucleotide diversity and reconstructing the phylogeny of related species due to its complexity" (lines 199-201). "Complexity" is a vague term. But chloroplast genomes are much smaller than nuclear genomes and thus can hardly be called more complex. I think, this sentence should be rephrased.
- You write that how chloroplast genomes are better for phylogenetic analysis than nuclear genomes (lines 199-211). However, the chloroplast genome is not always inherited in the same way as the nuclear genome. Phylogenies of chloroplasts and nuclei may differ. See, for example, https://pubmed.ncbi.nlm.nih.gov/24069353/ , https://pubmed.ncbi.nlm.nih.gov/28565145/ and https://pubmed.ncbi.nlm.nih.gov/24554473/ . I think this downside of using chloroplasts for phylogenetics is worth mentioning in the article. Since the nuclear genome has approximately 100 times more genes than the chloroplast genome, the nuclear phylogeny may be considered more important in some sense.
- The meaning of vertical grey lines in Figure S1 is not explained.
- Table S1. "Subunits of NADH-dehydrogenase". The complex that you mean is not a NADH-dehydrogenase. It was previously thought to be a NADH-dehydrogenase, but was later proved to have another function (https://pubmed.ncbi.nlm.nih.gov/26519774/). Thus, it's more correct to call it "NADH dehydrogenase-like complex" or just "NDH complex".
- Table S1. "Self-replication". Replication is when a copy of a molecule is created. Replication is synthesis of DNA based on DNA or synthesis of RNA based on RNA (as in viruses). See https://www.biologyonline.com/dictionary/replication and https://www.britannica.com/science/replication . Transcription and translation are not replication. Please, put "DNA dependent RNA polymerase" into a section named "Transcription". Also, please put "Large subunit of ribosome" and "Small subunit of ribosome" into a section named "Translation".
- Table S1. You put Ycf2 into the "Unknown" section. However, the function of Ycf2 is known already. It is a component of the plastid translocon (https://pubmed.ncbi.nlm.nih.gov/30309901/).
Author Response
- The are typos that should be fixed. "85 protein-coding130 genes" (line 102), "which were classified into fourfive categorie" (line 103), "have been submitted to the NCBI databankdatabas" (line 104).
Responses: We thank the reviewer for pointing out there errors, which has now been rectified. "85 protein-coding130 genes", "which were classified into fourfive categories" (line 103), "have been submitted to the NCBI databankdatabase" (line 104).
We checked the full manuscript again to avoid similar errors.
- In Figure 1, some gene symbols are truncated, like "rp..." or "r...". I think, it is better either to remove gene symbols, or write genes symbols for all 130 genes completely.
Responses: Figure 1 mainly shows an outline of the locations of genes, CDSs, inverted repeat regions and their structural features in the chloroplast genome of Lirianthe plants. Labeling the names of all 130 genes would be inconvenient for layout and reading, so the truncated gene names have been removed.
- In Figure 1, it is not clear what two green arrows in the rRNA row mean. rRNAs are denoted by red arrows there.
Responses: Thank you for pointing this out. The two green arrows in the rRNA row of Figure 1 were an output of the software automatically splitting overlapping genes into separate rows when generating the display results. We have now noticed this issue and have corrected it so that all rRNAs are denoted by red arrows in one row. Thank you for catching this oversight, we appreciate you helping improve the clarity of our figure.
- Please indicate in the caption of Figure 5 that the numbers above branches are bootstrap support values.
Responses: We have updated the caption of Figure 5 to indicate that the numbers above branches are bootstrap support values, as follows:
"Figure 5. Phylogenetic tree constructed using maximum likelihood method based on whole chloroplast genome sequences of the genus Lirianthe and other Magnoliaceae species. Numbers above branches are bootstrap support values based on 1000 replicates."
- "The chloroplast genome is a superior option to the nuclear genome for studying nucleotide diversity and reconstructing the phylogeny of related species due to its complexity" (lines 199-201). "Complexity" is a vague term. But chloroplast genomes are much smaller than nuclear genomes and thus can hardly be called more complex. I think, this sentence should be rephrased.
Responses: Thank you for the suggestion. You are right that using the term "complexity" here is inaccurate. We agree that the chloroplast genome is much smaller compared to the nuclear genome, so using "complexity" is inappropriate. Following your advice, we have removed the word “complexity” from use. We greatly appreciate your advice to help improve the accuracy of our paper.
- You write that how chloroplast genomes are better for phylogenetic analysis than nuclear genomes (lines 199-211). However, the chloroplast genome is not always inherited in the same way as the nuclear genome. Phylogenies of chloroplasts and nuclei may differ. See, for example, https://pubmed.ncbi.nlm.nih.gov/24069353/ , https://pubmed.ncbi.nlm.nih.gov/28565145/ and https://pubmed.ncbi.nlm.nih.gov/24554473/ . I think this downside of using chloroplasts for phylogenetics is worth mentioning in the article. Since the nuclear genome has approximately 100 times more genes than the chloroplast genome, the nuclear phylogeny may be considered more important in some sense.
Responses: Thank you for the excellent suggestion. You have rightly pointed out that using chloroplast genomes for phylogenetic analysis also has limitations, as the inheritance of chloroplast genomes can differ from that of nuclear genomes, which may lead to discrepancies between phylogenies constructed using the two genomes. We fully agree this is a very important point to make. The size of chloroplast genome of Arabidopsis thaliana (NCBI accession number: NC_000932.1) is 154,478 bp, while the one of nuclear genome is about 135 Mb (https://www.arabidopsis.org/portals/genAnnotation/gene_structural_annotation/agicomplete.jsp), with a haploid chromosome number of five. The website of phylogenetic relationships for flowering plants with genomes sequenced and published (https://www.plabipd.de/index.ep) listed four species in Magnoliaceae, with genome sizes of 1.75G (Liriodendron chinense), 2.24G (Magnolia biondii), 1.7G(Magnolia obovata) and 1.76G (Magnolia officinalis), respectively. Our research indicates that the chloroplast genome size of Lirianthe species in the family Magnoliaceae is approximately 160 kb. The size of plant nuclear genome is about 1,000~10,000 times larger than the chloroplast genome, which about 100~300 times more genes in nuclear genome than chloroplast one. The nuclear genome has far more genes than the chloroplast genome, and in some respects may provide more important information for phylogenies. Therefore, the ideal approach is to analyze the chloroplast and nuclear genomes together to provide a more comprehensive picture of phylogenetic relationships. After more than 20 years of development, molecular phylogeny studies have progressed from initially being based on single or several genes towards combining multiple genes, and even towards entire organellar genomes. But due to limitations such as cost, assembly and analytical techniques, it is still difficult to utilize the entire nuclear genome for systematic studies. The conventional approach is still to combine nuclear ribosomal RNA genes and their intergenic spacer regions from the nuclear genome, the most commonly used nuclear genome fragment is ITS. We also successfully assembled the ITS sequences of the eight Lirianthe species of Magnoliaceae in this study using GetOrganelle software. However, due to the lack of more ITS information for other Magnoliaceae species required for phylogenetic tree construction, we eventually gave up the analysis and interpretation of the ITS data in this paper. We hope to conduct related research in the future with collecting more Magnoliaceae plants and performing DNA sequencing.
- The meaning of vertical grey lines in Figure S1 is not explained.
Responses: Thank you for pointing out that the vertical grey lines in Figure S1 were not explained. The vertical grey lines indicate regions where differences occur in the sequence alignment results. The light grey background represents areas where the aligned sequences are completely identical.
We have updated the figure legend to clarify the meaning of the vertical grey lines.
- Table S1. "Subunits of NADH-dehydrogenase". The complex that you mean is not a NADH-dehydrogenase. It was previously thought to be a NADH-dehydrogenase, but was later proved to have another function (https://pubmed.ncbi.nlm.nih.gov/26519774/). Thus, it's more correct to call it "NADH dehydrogenase-like complex" or just "NDH complex".
Responses: Based on your suggestions, we have replaced "Subunits of NADH-dehydrogenase" by "NADH dehydrogenase-like complex".
- Table S1. "Self-replication". Replication is when a copy of a molecule is created. Replication is synthesis of DNA based on DNA or synthesis of RNA based on RNA (as in viruses). See https://www.biologyonline.com/dictionary/replication and https://www.britannica.com/science/replication . Transcription and translation are not replication. Please, put "DNA dependent RNA polymerase" into a section named "Transcription". Also, please put "Large subunit of ribosome" and "Small subunit of ribosome" into a section named "Translation".
Responses: To ensure accuracy, we will follow your suggestions to: 1) place "DNA dependent RNA polymerase" under a group titled "Transcription"; 2) Place "Large subunit of ribosome" and "Small subunit of ribosome" under a group titled "Translation".
We sincerely appreciate you reviewing our table closely and providing insightful suggestions for modification. This will help us enhance the precision of our wording. Thanks!
- Table S1. You put Ycf2 into the "Unknown" section. However, the function of Ycf2 is known already. It is a component of the plastid translocon (https://pubmed.ncbi.nlm.nih.gov/30309901/).
Responses: We place "ycf2" under a group titled "Preprotein translocation" follow your suggestion.
Reviewer 2 Report
Comments and Suggestions for Authors
The article titled "Comparative Analysis of the Chloroplast Genomes of Eight Species of the Genus Lirianthe Spach" focuses on sequencing and analyzing the chloroplast genomes of eight Lirianthe species. The study aims to understand their phylogenetic relationships within the family Magnoliaceae. The paper presents detailed information on genome sizes, gene content, codon usage, repeat sequence identification, DNA polymorphism, and phylogenetic analysis.
Major issues:
-
The study is simply descriptive, without clearly pointed purpose. If the main aim was to resolve phylogenetic relationships, the methodological approach should be revisited. The current version is under standard and is based only on ML analysis. What about other methods, genes trees, dataset partitioning? How interpret poorly resolved clades?
-
Few chapters present rather negative results - if there are no changes in IR content, there’s nothing to discuss. Also what was the purpose of SSR detection? It’s out of any context. The same situation with RSCU. Any evolutionary insights?
-
What about molecular delimitation based on complete plastomes? How many molecular diagnostic characters were found for each species? Nucleotide diversity is not always a good signature of a good barcode.
Minor issues:
Figure 5 is unreadable. Moreover, clades with poor bootstrap support (<70%) should be collapsed, since they are not statistically supported.
Figure. 4 should be removed, all are identical.
Table S3 is lacking information about sequence coverage of the newly assembled plastomes
The manuscript's primary focus on descriptive analysis, without a clearly defined purpose or application, significantly limits its scientific value. Future revisions should aim to articulate a clear research question or hypothesis and discuss potential applications and implications of the findings. Integrating analytical or experimental elements could also enhance the study's contribution to the field. Without these improvements, the manuscript remains a detailed yet ultimately limited exploration of the chloroplast genomes of Lirianthe species.
Author Response
Major issues:
1.The study is simply descriptive, without clearly pointed purpose. If the main aim was to resolve phylogenetic relationships, the methodological approach should be revisited. The current version is under standard and is based only on ML analysis. What about other methods, genes trees, dataset partitioning? How interpret poorly resolved clades?
Responses: Thank you very much for your constructive comments. We agree that the methodology of the paper needs to be strengthened. The main purpose of this study was to first describe the genome structure and characteristics after assembling and annotating the chloroplast genomes of 8 Lirianthe species, and then analyze the molecular phylogeny of Magnoliaceae using the whole chloroplast genome sequences and protein-coding gene sequences, with a focus on species delimitation within the genus Lirianthe . According to your suggestions, we have supplemented the content related to the phylogeny constructed using protein-coding gene sequences. Branches with support values lower than 70% have been interpreted. We recognize that an integrative application of multiple methods and a comprehensive analysis of different data are crucial for improving the reliability of phylogenetic interpretation. Thank you for your advice, which will help us adopt a more rigorous methodology to enhance the quality of the study.
2.Few chapters present rather negative results - if there are no changes in IR content, there’s nothing to discuss. Also what was the purpose of SSR detection? It’s out of any context. The same situation with RSCU. Any evolutionary insights?
Responses: Some sections in the manuscript present many "negative" results, such as almost no changes in inverted repeat regions, identical RSCU results - the relative synonymous codon usage of the eight species is consistent, and the analysis results of SSRs, etc. These are routine contents in the analysis of chloroplast genome structure and characteristics, and the slight interspecific differences precisely reflect the conservative evolution of Lirianthe species. Therefore, routine presentation is necessary, we just did not discuss these results in more detail in the results and discussion sections.
3.What about molecular delimitation based on complete plastomes? How many molecular diagnostic characters were found for each species? Nucleotide diversity is not always a good signature of a good barcode.
Responses: Based on your first suggestions and comments, we have supplemented the phylogenetic tree construction using protein coding genes in addition to using the whole chloroplast genome sequences. Both support our proposed delimitation of Lirianthe species - they are monophyletic in our proposed classification system, which is supported in the phylogenetic tree constructed using both datasets. While they are polyphyletic or paraphyletic in other classification systems. Please refer to the discussion section in the manuscript for details (line306-330).
The molecular diagnostic characteristic of a species, which we understand here as the DNA barcode to identify that species, is now considered by many studies to be the whole chloroplast genome as a super barcode. (10.1186/s13020-021-00460-z; 10.1371/journal.pone.0277809; 10.3389/fpls.2022.764255; 10.3389/fphar.2019.01441). Using several DNA fragments as barcodes for species identification is necessary in some scenarios, such as identification of protected species or authentication of medicinal materials. But these need to be determined at the population level of a certain species. We will carry out related work on identification of protected Magnoliaceae species in the future, after all, many Magnolia species are protected species.
Minor issues:
- Figure 5 is unreadable. Moreover, clades with poor bootstrap support (<70%) should be collapsed, since they are not statistically supported.
Responses: Through comparative analysis with the phylogenetic tree constructed using protein coding genes, we found that some branches in the phylogeny constructed using whole chloroplast genome sequences have lower support values, suggesting that the phylogenetic reconstruction method and dateset we used is not well-adapted or optimized. Here, we appreciate the your suggestion to increase different datasets to strengthen the research results. Your valuable comments have played a significant role in improving the quality of the manuscript.
- Figure. 4 should be removed, all are identical.
Responses: Thank you for your comments on Figure 4. We understand your concern about the high similarity of the IR boundaries across the eight species shown. Although the positions of rps19 and trnH near the IR boundaries are largely identical, there are minor differences that the figure captures visually. Our intention with this figure is to provide an intuitive illustration of the basic features of the IR regions and boundary genes. As such, we believe retaining the figure would be beneficial for readers to quickly grasp key information about the IRs.
- Table S3 is lacking information about sequence coverage of the newly assembled plastomes
Responses: We have supplemented the coverage evaluation results of the newly assembled chloroplast genomes in the first section\paragraph of the Results, and graphically presented the evaluation results, considering it as an important parameter for assessing the assembly quality of the chloroplast genomes. This will allow readers to assess the completeness and reliability of the genome assemblies.
- The manuscript's primary focus on descriptive analysis, without a clearly defined purpose or application, significantly limits its scientific value. Future revisions should aim to articulate a clear research question or hypothesis and discuss potential applications and implications of the findings. Integrating analytical or experimental elements could also enhance the study's contribution to the field. Without these improvements, the manuscript remains a detailed yet ultimately limited exploration of the chloroplast genomes of Lirianthe species.
Responses: We sincerely appreciate you taking the time to provide this insightful comments of our manuscript. According your recommendations, we have revised the manuscript to clearly defined the research purpose: 1) elucidate genomic structure, gene content and genetic variability encoded in Lirianthe chloroplast genome; 2) conduct the chloroplast phylogenomic analysis in the family Magnoliaceae using CPG sequences and protein coding sequence separately, with an emphasis of the generic delimitation for Lirianthe. Your suggestions significantly improve our manuscript by establishing a clear purpose, enhancing analytical rigor, and elucidating the broader scientific implications of our work. Thank you.
Round 2
Reviewer 2 Report
Comments and Suggestions for Authors
Dear authors,
Again, lack of any superbarcoding approach, but as far a I understand authors prefer to focus on phylogenetic relationships
Why do authors use different methods for whole plastome (RaxML) and PCG based (ML) trees? What's the reason behind it?
The analysis of relationships based on PCG is misinterpreted. Authors seem to not understand ultrafast bootstrap tests (Minh et al. 2013), since all values below 95 are simply insignificant and nodes with these values should be classified as not resolved. This severely impacts results, discussion and conclusion. I would recommend using at least two approaches for each of three: ML based and Bayesian inference. Also since in both trees clades support values are far from optimal, PartitionFinder and a posteriori partitioning analysis is strongly recommended. Each of the partitions can have different evolutionary models, but both IQ-trees and MrBayes support a multimodel approach.
Author Response
Using the whole plastomes as "superbarcodes" to aid identification and classification of closely related species has been applied in many studies. (10.1186/s13020-021-00460-z; 10.1186/1741-7007-7-84; 10.1186/s13059-016-1004-2) However, this is not the focus of our study. The main aims of this study were to annotate the newly assembly of chloroplast genomes for eight Lirianthe species, analyze their structural features and variations, and infer their phylogenetic relationships based on chloroplast genomic data, which we believe will contribute to the overall classification of Magnoliaceae plants.
In response to your concerns about the tree-building methods used in our phylogenetic analyses, which can significantly impact reliability of results, we have made the following revisions and improvements to the manuscript:
Firstly, we removed 2 spurious species, MN783014 (Lirianthe delavayi) and MW800876 (Houpoea rostrata) from the original dataset used in phylogenetic reconstruction, and added a new accession NC020316 (Houpoea officinalis). The spurious species analysis showed MN783014 and MW800876 had extremely long branch lengths (0.005860477 and 0.008228228 respectively), far exceeding the average branch length of 0.000282543 and the threshold of 0.005650862, so they were deleted. To ensure that each genus is represented by at least 2 species in the phylogenetic tree of Magnoliaceae, we replaced MW800876 with NC020316.
(Note: Spurious species were identified based on the criteria or threshold that their terminal branch lengths were at least 20 times longer than the average of all branch lengths across the whole tree. )
Then, we constructed phylogenetic trees using the whole chloroplast genome sequences (dataset: Wcp) of 44 Magnoliaceae species with both ML (Figure 1) and MrBayes (Figure 2) methods. The trees built by the two methods were highly congruent, and importantly, had higher Bootstrap values than trees from the original dataset.
Next, we used the concatenated alignment of 76 shared protein-coding genes (PCGs) from the 44 chloroplast genomes (dataset: PCGs) , and selected optimal partitioning models using MolderFinder. ML (Figure 3) and Bayesian trees (Figure 4) were built based on the partitioned models. The phylogenetic relationships resolved by the PCGs dataset were similar to those inferred from the Wcp dataset, with over 85% ~ 90% of nodes having Bootstrap values above 95%.
Finally, we revised text contexts in the Methods, Results and Discussion sections related to phylogenetic analysis accordingly.
We believe these analyses and revisions have produced more robust phylogenetic trees and strengthened the manuscript. Thank you again for your constructive suggestions.

Round 3
Reviewer 2 Report
Comments and Suggestions for Authors
Some minor comments to the latest revision.
Methods section:
There’s no Mybayes program but MrBayes. Please provide the version number too.
How was the optima burn-in generations estimated?
The ML analysis description is a real mess. Once authors claim they used RAxML, the other time Iq-tree. If RAxML was used for WCP dataset, why Iqtree was used for PCG?
It’s hard to follow, and this paragraph has to be rewritten. Why 10,000 Ubs were applied?
Does the Gblocks apply to the WCP dataset too? Do the partitions apply to this dataset too?
To simplify, I would remove PCG dataset from this paper, since it’s far less informative and as far as I know, there’s no conflicts between the PCG and wcp datasets
Figure 4. I recommend moving it to supplementary material, it doesn’t bring any significant content, all species have almost identical plastome structure.
Figure 5. I would present only BS values lower than 100 for better readability
Figure 6. I would leave only PP values lower than 1. These values are not bootstraps as stated in the caption, but posterior probabilities
Supplementary figure 1. There are errors on coverage estimates - probably in the case of F,G and H too low restriction were applied and the regions of rRNA genes were contaminated (by nuclear or mt rRNA) or doubled mapped (since they are at IRs).
Author Response
Methods section: There’s no Mybayes program but MrBayes. Please provide the version number too.
Response: The spelling mistake has been corrected. The version number of MrBayes is 3.2.6, which has been added in the revised manuscript.
How was the optima burn-in generations estimated?
Response: When MrBayes starts running, the“Progress”group box will show the“Average standard deviation of split frequencies”(ASDSF) value in real time. In general, when the ASDSF value falls below 0.01, it can be used as an indicator that the Bayesian inference (BI) run has convergent.
The ML analysis description is a real mess. Once authors claim they used RAxML, the other time Iq-tree. If RAxML was used for WCP dataset, why Iqtree was used for PCG?
Response: The WCP dataset was not partitioned and model optimized, while the PCG dataset was partitioned and model optimized. Therefore, the former was analyzed using the RAxML program, while the latter was analyzed using the IQ-TREE program. We conduct data analysis based on our familiarity with Geneious, PhyloSuite software, and their Plugins programs. Based on your suggestions below, we have removed the PCG dataset to provide a more concise and clear description of the analysis methods and results for the WCP dataset.
It’s hard to follow, and this paragraph has to be rewritten. Why 10,000 Ubs were applied?
Response: We executed bootstrap 1,000 and 10,000 times, and the support value for bootstrap obtained from 10,000 times was not higher than that from 1,000 times. So, we will still set the bootstrap count to the commonly used 1,000 times in the revised manuscript.
Does the Gblocks apply to the WCP dataset too? Do the partitions apply to this dataset too?
Response: The Gblocks program can be applied to WCP datasets as long as its data type is set to "nucleotide". Because the partitioning mode requires the structural information, such as protein coding regions, rRNA regions, tRNA regions, and gene spacer regions, which we did not attach or add into analysis, so the WCP dataset is not suitable for partitioning processing.
To simplify, I would remove PCG dataset from this paper, since it’s far less informative and as far as I know, there’s no conflicts between the PCG and wcp datasets
Response: The phylogenetic tree results constructed from the PCG dataset and the WCP dataset are consistent. We accept your suggestion and have removed the analysis methods and results of the PCG dataset in the revised manuscript, retaining only the analysis content of the WCP dataset.
Figure 4. I recommend moving it to supplementary material, it doesn’t bring any significant content, all species have almost identical plastome structure.
Response: We accept your suggestion and have moved Figure 4 to the supplementary file in the revised manuscript.
Figure 5. I would present only BS values lower than 100 for better readability
Response: We have re-presented the diagram with only BS values lower than 100 based on your suggestion.
Figure 6. I would leave only PP values lower than 1. These values are not bootstraps as stated in the caption, but posterior probabilities
Response: Same as above. Revised.
Supplementary figure 1. There are errors on coverage estimates - probably in the case of F,G and H too low restriction were applied and the regions of rRNA genes were contaminated (by nuclear or mt rRNA) or doubled mapped (since they are at IRs).
Response: Firstly, when mapping the raw sequencing data to the assembled chloroplast genome, we initially used minimap for mapping. In response to your concerns, we have used bwa and bowtie2 separately for mapping, and the results still show that the coverage of the regions is about two or three times as high as other areas (Figure 1). Even though we filtered the reads using the samtools after alignment by bowtie2 (specific command: samtools view - h file_aligned. bam | grep - v "XA: Z:" | samtools view - bS ->file_aligned_filtered. bam), only the reads that were mapped exactly 1 time were retained, and the results remained unchanged.
Secondly, we compared the cp genomes of these three "questionable" species with those of the other five "normal" species, and carefully examined the higher coverage regions (103,500~111,500; 135,000~143,000), but did not find significant differences in these regions.
Thirdly, in order to verify whether it was possible that the chloroplast genome had transferred to the nuclear or mitochondrion genome, we downloaded the nuclear genome (GCA_021015155.1_ASM2101515v1_genomic.fna, 1.7 Gb) and mitochondrion genome (NC064401, 930306 bp) of Magnolia officinalis (also named Houpoea officinalis), and then aligned the chloroplast genome of the "questionable" species with the nuclear and mitochondrion genome of Magnolia officinalis through blastn analysis. Indeed, we found highly similar regions between them (Table 1 and Table 2). Due to the lack of the nuclear and mitochondrial genomes of the "questionable" species themselves, only the ones of the closely related species, Magnolia officinalis, can be used here. So, we currently cannot verify whether the "questionable" species themselves had undergone DNA transfer.
However, based on the results of the mapping analysis using minimap, bwa, and bowtie2, it can be concluded that the coverage analyses for these three "questionable" species are consistent. Overall, we present the coverage results here only to demonstrate that the assembly quality of the eight Lirianthe species’s chloroplast genome is guaranteed, and whether there is DNA transfer between the chloroplast genome and nuclear genome or mitochondrial genome does not affect our assembly quality. So supplementary figure S1 remains unchanged.
